

# The preventive effect of dexmedetomidine on paroxysmal sympathetic hyperactivity in severe traumatic brain injury patients who have undergone surgery: a retrospective study

Qilin Tang[1,2], Xiang Wu[1,2], Weiji Weng[1,2], Hongpeng Li[3], Junfeng Feng[1,2], Qing Mao[1,2], Guoyi Gao[1,2] and Jiyao Jiang[1,2]

[1] Department of Neurosurgery, Ren Ji Hospital, School of Medicine, Shanghai Jiao Tong University, Shanghai, China
[2] Shanghai Institute of Head Trauma, Shanghai, China
[3] Department of Neurosurgery, Rizhao City Hospital of Traditional Chinese Medicine, Rizhao, Shandong Province, China

## ABSTRACT

**Background.** Paroxysmal sympathetic hyperactivity (PSH) results and aggravates in secondary brain injury, which seriously affects the prognosis of severe traumatic brain injury patients. Although several studies have focused on the treatment of PSH, few have concentrated on its prevention.

**Methods.** Ninety post-operation (post-op) severe traumatic brain injury (sTBI) patients admitted from October 2014 to April 2016 were chosen to participate in this study. Fifty of the post-op sTBI patients were sedated with dexmedetomidine and were referred as the "dexmedetomidine group" (admitted from May 2015 to April 2016). The other 40 patients (admitted from October 2014 to May 2015) received other sedations and were referred as the "control group." The two groups were then compared based on their PSH scores and the scores and ratios of those patients who met the criteria of "probable," "possible" and "unlikely" using the PSH assessment measure (PSH-AM) designed by *Baguley et al. (2014)*. The durations of the neurosurgery intensive care unit (NICU) and hospital stays and the Glasgow outcome scale (GOS) values for the two groups were also compared to evaluate the therapeutic effects and the patients' prognosis.

**Results.** The overall PSH score for the dexmedetomidine group was $5.26 \pm 4.66$, compared with $8.58 \pm 8.09$ for the control group. The difference between the two groups' PSH scores was significant ($P = 0.017$). The score of the patients who met the criterion of "probable" was $18.33 \pm 1.53$ in the dexmedetomidine group and $22.63 \pm 2.97$ in the control group, and the difference was statistically significant ($P = 0.045$). The ratio of patients who were classified as "unlikely" between the two groups was statistically significant ($P = 0.028$); that is, 42 (84%) in the dexmedetomidine group and 25 (62.5%) in the control group. The differences in NICU, hospital stays and GOS values between the two groups were not significant.

**Conclusion.** Dexmedetomidine has a preventive effect on PSH in sTBI patients who have undergone surgery.

Corresponding author
Guoyi Gao, gao3@sina.com

## INTRODUCTION

Sympathetic activity after stress is the body's necessary protective response, but sympathetic overactivity following acute brain injury fosters hemodynamic instability and contributes to secondary brain damage, which severely affects the prognosis (*Baguley et al., 2006*; *Hinson & Sheth, 2012*; *Lv et al., 2011*). Paroxysmal sympathetic hyperactivity (PSH) is a syndrome characterized by paroxysmal episodes of sympathetic surges that can manifest as hyperthermia, diaphoresis, tachycardia, hypertension, tachypnea, and dystonic posturing (*Baguley et al., 2014*). Since Wilder Penfield's first report on the syndrome, numerous terms have been used to describe it, including diencephalic autonomic epilepsy, dysautonomia, paroxysmal autonomic instability with dystonia, and PSH (*Penfield, 1929*; *Perkes et al., 2011*). It has been reported that 7.7–33% of traumatic brain injury patients in intensive care units (ICUs) suffer from this disorder (*Baguley et al., 2007b*; *Dolce et al., 2008*; *Fearnside et al., 1993*; *Fernandez-Ortega et al., 2012*; *Fernandez-Ortega et al., 2006*; *Hendricks et al., 2007*; *Lv et al., 2010*; *Lv et al., 2011*; *Perkes et al., 2010*; *Perkes et al., 2011*; *Rabinstein, 2007*).

PSH episodes can last for several minutes to hours and may recur multiple times during the day (*Blackman et al., 2004*). Unstable conditions such as high blood pressure or fever, can result in and aggravate secondary brain injury, which is considered to be one of the main causes of unfavorable prognosis (*Baguley, 2008a*; *Baguley et al., 2007a*; *Fernandez-Ortega et al., 2012*; *Fernandez-Ortega et al., 2006*; *Greer et al., 2008*). PSH can also cause a hypermetabolic state, unopposed inflammation, and weight loss–all of which can lead to worse outcomes and prolonged hospital stays (*Choi et al., 2013*; *Mehta et al., 2008*; *Tracey, 2007*). Several studies have focused on how to treat PSH, but few have concentrated on its prevention.

Dexmedetomidine is a new selective alpha-2 agonist that has been shown to decrease sympathetic activation, with the effect on sedation, analgesia, and antianxiety without significant inhibition of respiration (*Martin et al., 2003*; *Panzer, Moitra & Sladen, 2011*; *Venn, Hell & Grounds, 2000*). It has been formally approved by the Food and Drug Administration of the United States of America (FDA) for use in sedating ICU patients since 1999. The Department of Neurosurgery at Ren Ji Hospital introduced dexmedetomidine for the routine postoperative sedation of sTBI patients in May 2015. In this article, we report that dexmedetomidine may have a preventive effect on PSH in postoperative patients who have suffered severe traumatic brain injury.

## METHODS

### Study population

Retrospectively and consecutively, all of the patients included in this study were admitted to the Department of Neurosurgery at Ren Ji Hospital from October 2014 to April 2016. Consecutive patients were included if they were older than 18, had suffered sTBI, and had undergone an operation. Patients were excluded if they met any of the following criteria:

serious hepatic dysfunction (Child-Pugh class B or C); serious renal dysfunction (undergoing dialysis before surgery; or serum creatinine >445 μmol/L and/or blood urea nitrogen >20 mmol/L in preoperative laboratory examination); unstable haemodynamics at NICU admission, such as bradycardia and hypotension; death occurring within 24 h after surgery. Of the 103 consecutively recorded patients, 13 were excluded based on the preceding criteria. Based on the post-op sedation received, the remaining 90 patients were divided into two groups: the control group (admitted from October 2014 to May 2015) and the dexmedetomidine group (admitted from May 2015 to April 2016). This study was approved by the Ethics Committee of Ren Ji Hospital (Ethical Application Ref: [2016]W018).

## Treatment protocol

For the dexmedetomidine group, when each patient was placed in the NICU after his or her operation, dexmedetomidine was administrated at an initial loading dose of 0.8 μg/kg within 10 min, followed by a continuous infusion at 0.25–0.75 μg/(kg h)$^{-1}$. For the control group, propofol or midazolam was administered by intravenous pump infusion. Propofol was initiated at 2 mg/(kg h)$^{-1}$ and midazolam was initiated at 0.1 mg/(kg h)$^{-1}$. Diazepam is added at the discretion of doctors as a component of a goal-directed sedation regimen. All of the patients were maintained at a Riker sedation-agitation scale (SAS) of 3–4 and the sedations were withdrawn gradually starting on day 5, depending on the patient's condition. Upon the onset of PSH symptoms, symptomatic treatments such as cooling the temperature and lowering the heart rate were administrated as soon as possible. Verbal informed consent was obtained from patients' decision makers.

## Data collection and outcome assessment

In the NICU, on an hourly basis, the nurses recorded each patient's heart rate, respiratory rate, systolic blood pressure, temperature, sweating, and posture during episodes. Once the patients were transferred to ordinary wards, abnormal signs and symptoms of PSH were observed by trained nurses and events were reported to the doctors in charge of the patients and recorded. The patients were scored using the PSH Assessment Measure (PSH-AM) developed by *Baguley et al. (2014)*. The PSH-AM is designed to have two components–one addressing the probability of the diagnosis (the Diagnosis Likelihood Tool [DLT]) and another assessing the severity of the clinical features (the Clinical Feature Scale [CFS]). The numerical output of these two components are added together to estimate the diagnostic likelihood of PSH. Regarding the probability of a PSH diagnosis, a score of less than 8 indicated "unlikely," a score from 8 to 16 indicated "possible", and a score greater than or equal to 17 indicated "probable" (*Baguley et al., 2014*).

All patients received a three-month follow-up evaluation by telephone. The outcomes were quantified using the Glasgow Outcome Scale (GOS) (*Jennett & Bond, 1975*) at discharge and after three months. The GOS were evaluated based on the descriptions provided by the patients' relatives three months after injury to evaluate the prognosis. The lengths of the patients' NICU and hospital stays were also collected to assess the therapeutic effects.

## Statistical analysis

A statistical analysis was performed using SPSS Version 20.0.0 (SPSS Inc., Chicago, Illinois). The continuous variables were presented as mean ± standard deviation (SD) and the proportions were calculated for the categorical variables. For the continuous variables, group comparisons were made using parametric $t$-tests if the data followed normal distribution. Otherwise, non-parametric Mann–Whitney-$U$ tests were used. For the categorical variables, such as gender, diagnosis and surgery type, group comparisons were made using a chi-quadrat test or a Fisher's exact test if at least one value was <5. Significance was set at $p < 0.05$. To assess the power of our study, we conducted power calculations by using G*power (*Faul et al., 2007*).

## RESULTS

The data entry was completed in July 2016, when all of the patients had received their three-month follow-up. The patients' data were collected for retrospective analysis. No significant differences were noted between the two groups in relation to age, gender, pre-operative GCS, diagnosis, time from injury to surgery, and type of surgery. The demographic patient data are listed in Table 1.

The patients in the dexmedetomidine group were sedated for a period of 5.46 ± 2.82 days, compared with 6.08 ± 2.95 in the control group ($p = 0.317$). The total dose of dexmedetomidine given to each person was 4.79 ± 2.47 mg. Dexmedetomidine was not administrated to the patients in the control group. The PSH-AM score of the patients in the dexmedetomidine group was significantly lower than that of those in the control group (5.26 ± 4.66 vs. 8.58 ± 8.09, $P = 0.017$).

No significant differences were noted between the two groups regarding the ratio of patients who met the criteria of "probable" (three (6%) in the dexmedetomidine group vs. eight (12.5%) in the control group, $P = 0.056$) and "unlikely" (3.67 ± 2.40 in the dexmedetomidine group vs. 3.28 ± 2.30 in the control group, $P = 0.519$). However, the score of patients who met the "probable" criterion in the dexmedetomidine group was 18.33 ± 1.53, compared with 22.63 ± 2.97 in the control group ($P = 0.045$). Moreover, 42 (84%) of the patients in the dexmedetomidine group met the "unlikely" criterion, compared with 25 (62.5%) patients in the control group ($P = 0.028$). No significant differences in the groups were established for patients who met the "possible" criterion, in number or score (Figs. 1 and 2).

In NICU, the two groups had the same mortality of 20% (10 of 50 in the dexmedetomidine group and 8 of 40 in the control group). No more patients died after transfered to ordinary wards. The patients in the dexmedetomidine group stayed in the NICU for an average duration of 15.70 ± 13.07 days and in the hospital for 23.50 ± 16.58 days, compared with 20.65 ± 16.74 and 28.53 ± 20.28 days, respectively, in the control group ($P = 0.119$ and $P = 0.174$) (Fig. 3). The GOS value at discharge was 3.00 ± 1.28 in the dexmedetomidine group, compared with 2.75 ± 1.15 in the control group ($P = 0.338$). The GOS vaule three months after sTBI in the dexmedetomidine group was 3.42 ± 1.47, compared with 3.05 ± 1.43 in the control group ($P = 0.234$) (Fig. 4 and Table 1). The statistical power for the outcome measures described above is relatively low (21%–42%).

**Table 1  Baseline demographic, clinical characteristics, PSH-AM score and outcome of the study population.** Patients' age, gender, pre-operative GCS, and diagnosis, time from injury to surgery, type of surgery, day of sedation, the ratio and the PSH-AM score of the patients who met the criteria of "probable," "probable" and "unlikely," the NICU and Hospital stays, and GOS at discharge and after three months.

| Variables | Total | Dexmedetomidine group | Control group | p vaule |
|---|---|---|---|---|
| Demographic | | | | |
| No. of patients | 90 | 50 | 40 | – |
| Male gender | 62 (68.9%) | 37 (74%) | 25 (62.5%) | 0.242 |
| Age, mean ± SD, y | 46.76 ± 15.41 | 47.50 ± 15.12 | 45.83 ± 15.92 | 0.611 |
| Preoperative GCS, mean ± SD | 6.26 ± 1.77 | 6.24 ± 1.88 | 6.28 ± 1.65 | 0.926 |
| Diagnosis | | | | |
| Cerebral contusion without hemorrhage | 23 (25.6%) | 14 (28%) | 9 (22.5%) | 0.552 |
| Hemorrhagic cerebral contusion | 54 (60%) | 30 (60%) | 24 (60%) | 1 |
| Acute subdural hematoma | 72 (80%) | 38 (76%) | 34 (85%) | 0.289 |
| Acute epidural hematoma | 35 (38.9%) | 19 (38%) | 16 (40%) | 0.847 |
| Skull fracture | 65 (72.2%) | 36 (72%) | 29 (72%) | 0.958 |
| Subarachnoid hemorrhage | 78 (86.7%) | 43 (86%) | 35 (87%) | 0.835 |
| Cerebral hernia | 30 (33.3%) | 14 (28%) | 16 (40%) | 0.23 |
| Time from injury to surgery | 5.21 ± 1.75 | 5.03 ± 1.76 | 5.43 ± 1.71 | 0.89 |
| Type of surgery | | | | |
| Decompressive craniectomy | 87 (96.7%) | 48 (96%) | 39 (97.5%) | 1 |
| Epidural hematoma clearing | 35 (38.9%) | 19 (38%) | 16 (40%) | 0.847 |
| Subdural hematoma clearing | 72 (80%) | 38 (76%) | 34 (85%) | 0.289 |
| Hemorrhagic contusion clearing | 47 (52.2%) | 25 (50%) | 22 (55%) | 0.637 |
| Lateral ventriculopuncture drainage | 90 (100%) | 50 (100%) | 40 (100%) | 1 |
| Day of sedation | 5.73 ± 288 | 5.46 ± 282 | 6.08 ± 2.95 | 0.317 |
| PSH diagnostic likelihood | 6.73 ± 6.58 | 5.26 ± 4.66 | 8.58 ± 8.09 | 0.017 |
| Probable (≥17) | | | | |
| No. | 11 (12.2%) | 3 (6%) | 8 (20.0%) | 0.056 |
| Score | 21.45 ± 3.27 | 18.33 ± 1.53 | 22.63 ± 2.97 | 0.045 |
| Possible (8–16) | | | | |
| No. | 12 (13.3%) | 5 (10%) | 7 (17.5%) | 0.358 |
| Score | 11.17 ± 2.69 | 10.80 ± 3.56 | 11.43 ± 2.15 | 0.710 |
| Unlikely (<8) | | | | |
| No. | 67 (74.4%) | 42 (84%) | 25 (62.5%) | 0.028 |
| Score | 3.52 ± 2.35 | 3.67 ± 2.40 | 3.28 ± 2.30 | 0.519 |
| NICU stay, mean ± SD, d | 17.90 ± 14.93 | 15.70 ± 13.07 | 20.65 ± 16.74 | 0.119 |
| Hospital stay, mean ± SD, d | 25.87 ± 18.40 | 23.50 ± 16.58 | 28.53 ± 20.28 | 0.174 |
| GOS at discharge | 2.89 ± 1.22 | 3.00 ± 1.28 | 2.75 ± 1.15 | 0.338 |
| GOS after three months | 3.25 ± 1.46 | 3.42 ± 1.47 | 3.05 ± 1.43 | 0.234 |

**Notes.**

P values for differences between two treatment groups by Student t test or Fisher's exact test.

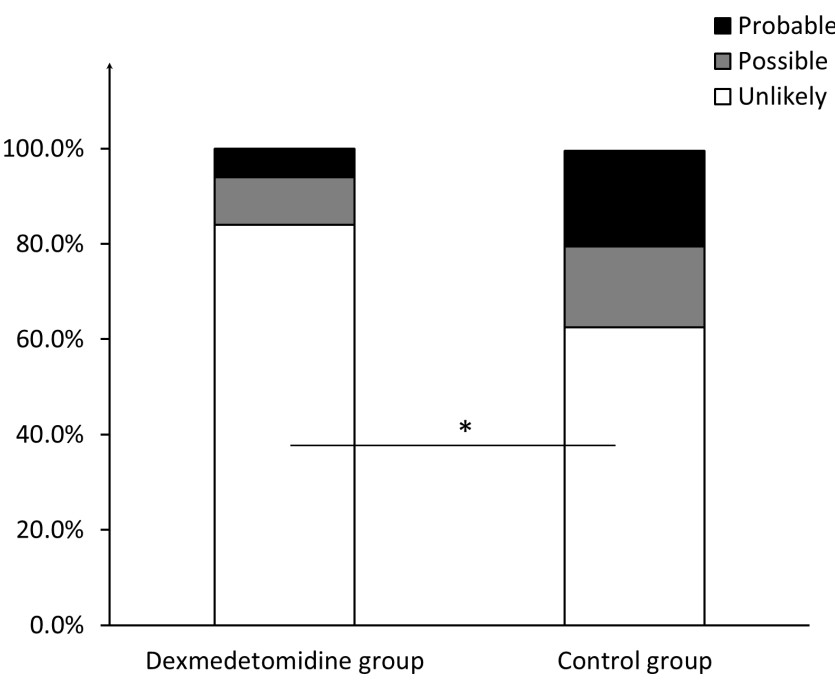

**Figure 1** **Ratio of patients meeting different criteria regarding likelihood of PSH diagnosis.** The ratio of patients who meet the criteria of "probable", "possible", and "unlikely" respectively, in the dexmedetomidine group and in the control group. * indicates significance ($p = 0.028$).

## DISCUSSION

Severe traumatic brain injury, which is associated with considerable mortality and morbidity, represents a significant public health problem around the world. PSH after sTBI is one of the important factors contributing to adverse outcomes. The first diagnostic criteria for PSH were published in 1993 (*Fearnside et al., 1993*). Since that time, different criteria have been proposed mainly due to the presence of the following signs and symptoms in the absence of other potential causes: fever, tachycardia, hypertension, tachypnea, excessive diaphoresis and extensor posturing, or severe dystonia (*Baguley et al., 2014*; *Blackman et al., 2004*; *Dolce et al., 2008*; *Perkes et al., 2011*; *Rabinstein, 2007*). A PSH diagnosis is one of exclusion. Considering differential diagnoses such as sepsis, systemic inflammatory response syndrom, or sedation withdrawal is crucial, but sometimes difficult. *Baguley et al. (2014)* proposed a consensus of diagnostic criteria for PSH–specifically, a probabilistic system that assigned a diagnostic likelihood rather than providing a definitive diagnosis. This diagnostic system enables medical workers to diagnose PSH more precisely and provides a quantization foundation for PSH evaluation.

The pathophysiology of PSH is unclear, and related theories have not been empirically tested. Initially, the cause of the condition was thought to be diencephalic discharges (*Bhigjee, Ames & Rutherford, 1985*; *Penfield, 1929*). However, later studies identified no seizure activities in PSH using electroencephalography (*Baguley et al., 2006*; *Boeve et al., 1998*). The current consensus is that epilepsy is not the cause of PSH (*Bullard, 1987*; *Pranzatelli, Pavlakis & Gould, 1991*; *Thorley, Wertsch & Klingbeil, 2001*). The accepted model is the

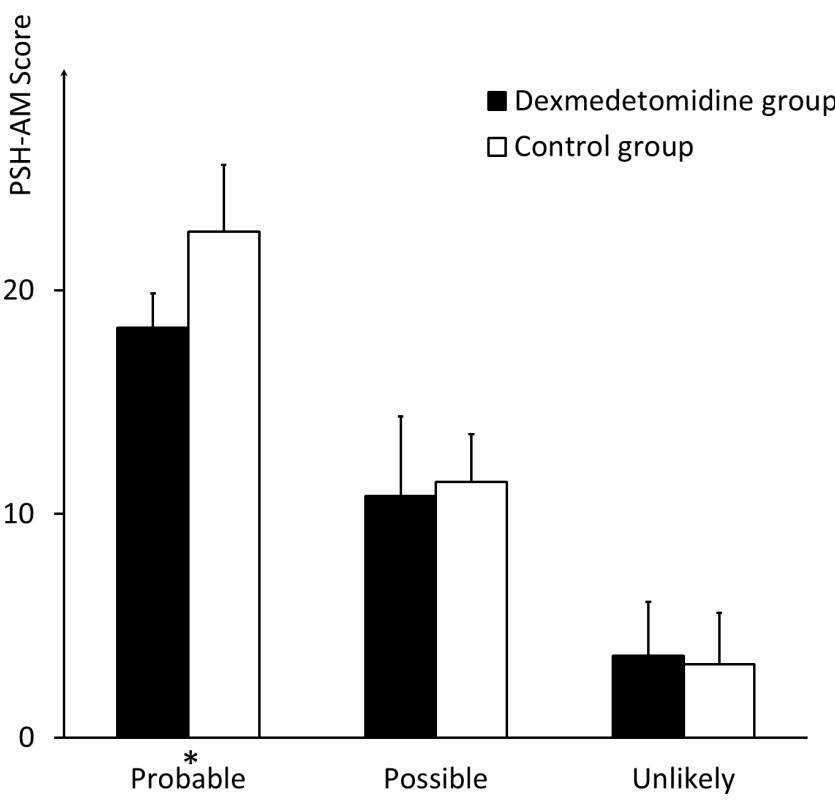

**Figure 2 Scores of patients meeting different criteria regarding likelihood of PSH diagnosis.** The comparison of the PSH scores of the patients who meet the criteria of "probable," "possible," and "unlikely," respectively, between the dexmedetomidine group and the control group. * indicates significance ($p = 0.045$).

excitatory–inhibitory ratio (EIR) model (*Baguley, 2008b*), which proposes that the afferent stimulus is normally controlled by tonic inhibitory drive from diencephalic centers. Once the tonic inhibition cycle is broken, there is a positive-feedback loop that produces sympathetic over-activity following any afferent stimuli (*Baguley et al., 2009a*; *Baguley et al., 2009b*). This model explains how a normally non-noxious stimulus can cause an uncontrolled sympathetic response.

For PSH, timely diagnosis, swift episode control, and reduced onset frequency are all crucial in improving prognosis. Regarding the treatment of PSH, beta-blockers, which attenuate sympathetic activation, are now widely used to control its onset (*Do, Sheen & Bromfield, 2000*; *Rabinstein & Benarroch, 2008*; *Sneed, 1995*). Morphine (a potent $\mu$-opioid receptor agonist), Bromocriptine (a dopamine receptor agonist), and baclofen (a GABA receptor agonist) have also been reported to successfully alleviate PSH episodes (*Becker et al., 2000*; *Cuny, Richer & Castel, 2001*; *Ko et al., 2010*; *Russo & O'Flaherty, 2000*). However, as few cohort studies guiding PSH treatment exist and the underlying pathophysiology remains unclear, treatment strategies frequently focus on controlling symptoms. For instance, antipyretics are administered to treat hyperthermia, sedatives for agitation, and antihypertensive medications for hypertension (*Choi et al., 2013*).

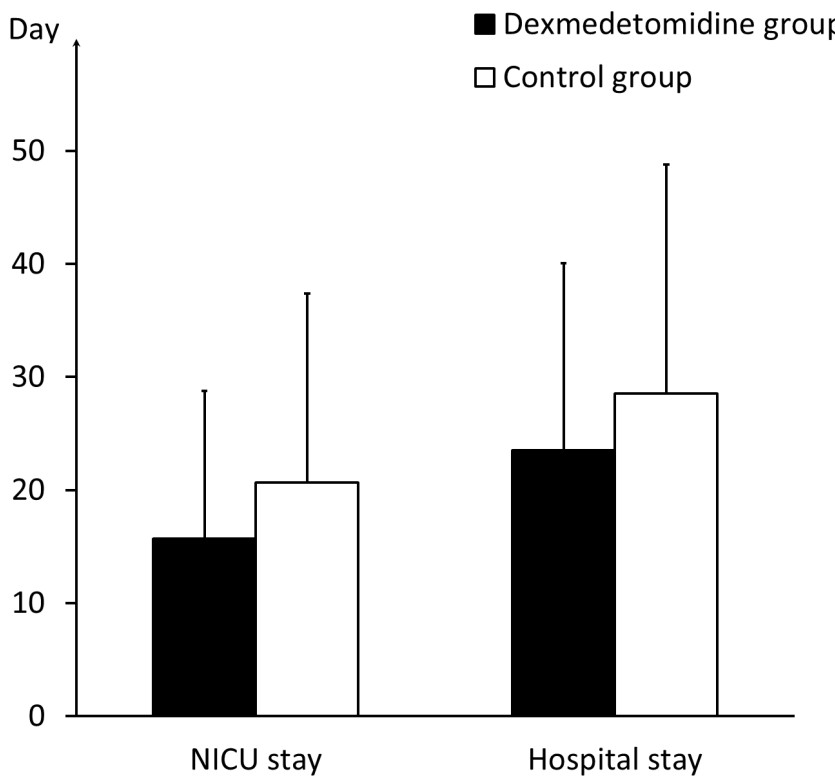

**Figure 3 NICU and hospital stay of the two groups.** The comparison of the duration of the NICU and hospital stays between the dexmedetomidine group and the control group.

Dexmedetomidine is a highly selective presynaptic $\alpha$-2 adrenergic agonist that may block norepinephrine release and enhance sympathetic inhibition to maintain the balance of the unregulated sympathetic feedback loop. Two cases of dexmedetomidine being used to diminish PSH symptoms have been reported (*Goddeau, Silverman & Sims, 2007*; *Kern et al., 2016*), which may support the disconnection theory. Unlike a traditional sedative such as propofol, which acts on GABA receptors in the cortex, dexmedetomidine displays the actions of analgesia, sedation, and anxiety treatment by acting on the $\alpha2$ adrenergic receptors in the locus coeruleus (LC) (*Nelson et al., 2003*). Patients sedated using dexmedetomidine can be awakened at any time to judge changes in their conscious state. It has also been reported that dexmedetomidine has a neuroprotective effect because it inhibits the apoptosis of nerve cells, protecting against local ischemia and slowing the progression of infarction (*Cai et al., 2014*; *Cosar et al., 2009*; *Dahmani et al., 2005*). These features make dexmedetomidine a commonly prescribed sedative in NICUs.

PSH prevention to decrease the number and severity of episodes has more significance in improving the outcome of sTBI, compared with the elimination of PSH episodes. Although several studies have focused on treating PSH, few have addressed its prevention. Thus, based on the routine use of dexmedetomidine for post-op sedation and data collected from patients, this study explored whether dexmedetomidine, which is reported to successfully eliminate PSH episodes, also has a preventive effect on this syndrome.

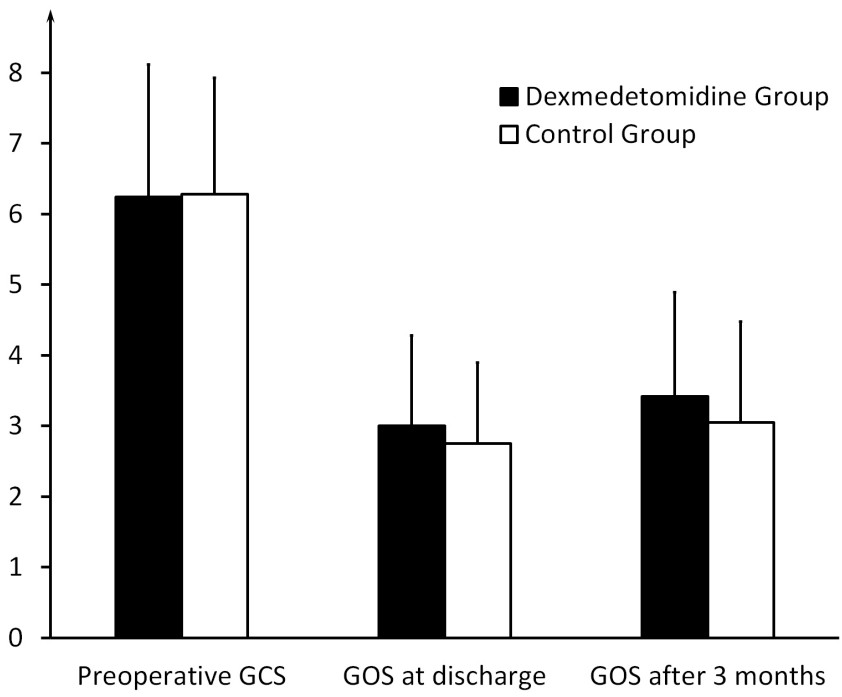

**Figure 4 Outcome of the two groups.** The preoperative GCS, GOS at discharge, and GOS after three months in the dexmedetomidine group and the control group.

We compared the overall PSH-AM scores between the two groups and found that the score for the dexmedetomidine group was significantly lower than that for the control group, indicating that the patients in the dexmedetomidine group had a lower probability of PSH diagnosis. Compared with the control group, the dexmedetomidine group had a lower score of patients meeting the "probable" criterion and a larger ratio of patients meeting the "unlikely" criterion, with a statistically significant difference. The preceding results showed that dexmedetomidine, to some extent, prevented the patients in the dexmedetomidine group from the onset of PSH.

The differences in duration of the NICU and hospital stays and in the GOS values at three-month follow-up between the two groups did not reach statistical significance. sTBI is a heterogeneous disease and patients' post-op conditions are variable and protean. Many factors may contribute to an sTBI prognosis. Moreover, the statistical power for the above three outcome measures is relatively low (21%–42%), which may be due to the small sample size. The low statistical power indicates low ability to distinguish the effect from random chance. This may be why our study did not reach a statistically significant difference for these three outcome measures.

The onset of PSH is often sudden and short in duration. In addition, the symptoms are often atypical and changeful, which can make a timely diagnosis difficult. Delayed diagnoses and treatment difficulties can result in prolonged NICU and hospital stays, higher medical costs, and poorer outcomes (*Hinson & Sheth, 2012*; *Lv et al., 2011*). Dexmedetomidine is a commonly used sedative in the NICU, and had a preventive effect on post-op PSH in

patients with sTBI who have undergone surgery in our study, although the exact mechanism remains unclear. This feature makes it a promising medication for the sedation of post-op sTBI patients and for preventing PSH onset, both of which are significant in reducing postoperative complications, cutting down the duration of hospitalization, and improving prognosis.

This study had several limitations. First, it was a single-center study, and thus the findings may not be generalizable to other centers. The two groups were studied over different periods, during which there might have been differences aside from the sedative used after surgery which were failed to recognize. Second, only those sTBI patients who underwent surgery were studied, and therefore these findings may not be generalizable to those who have not undergone surgery. Third, we observed only PSH episodes during hospital stays without long-term follow-up after discharge. Fourth, limited by objective conditions we evaluated GOS after three months based on the descriptions provided by the patients' relatives or health care givers, which might cause bias. And last, the statistical power for this study is relatively low (21%–42%), which indicates low ability to distinguish the differences. The preceding limitations impair, to some extent, the trustworthiness of our conclusions.

## CONCLUSIONS

This study showed that dexmedetomidine had a preventive effect on PSH in patients with sTBI who have undergone surgery. The effect was detected as compared to other sedation regimes (midazolam, propofol, and diazepam) with limited statistical power. To our knowledge, no randomized controlled trial to date has studied the preventive effect of dexmedetomidine on PSH for paients with sTBI. This must be confirmed by additional high-level, evidence-based medical research.

## ACKNOWLEDGEMENTS

The authors thank the nurses of the NICU and Neurosurgery Departments of Ren Ji Hospital for their active participation in the observing of PSH events.

### Funding
This work was supported by the National Natural Science Foundation of China (No. 81671198) and grants from the Shanghai Municipal Education Commission—Gaofeng Clinical Medicine Grant Support (No. 20152212). The funders had no role in study design, data collection and analysis, decision to publish, or preparation of the manuscript.

### Grant Disclosures
The following grant information was disclosed by the authors:
National Natural Science Foundation of China: 81671198.
Shanghai Municipal Education Commission—Gaofeng Clinical Medicine Grant Support: 20152212.

## Competing Interests

The authors declare there are no competing interests.

## Author Contributions

- Qilin Tang conceived and designed the experiments, analyzed the data, contributed reagents/materials/analysis tools, wrote the paper, prepared figures and/or tables, reviewed drafts of the paper.
- Xiang Wu, Weiji Weng and Hongpeng Li contributed reagents/materials/analysis tools, reviewed drafts of the paper.
- Junfeng Feng, Qing Mao, Guoyi Gao and Jiyao Jiang conceived and designed the experiments, reviewed drafts of the paper.

## Human Ethics

The following information was supplied relating to ethical approvals (i.e., approving body and any reference numbers):

This study was approved by the Ethics Committee of Renji Hospital (Ethical Application Ref: [2016]W018). Verbal consent was obtained from the patients' decision makers.

## Data Availability

The raw data has been supplied as Data S1.

## Supplemental Information

Supplemental information for this article can be found online at http://dx.doi.org/10.7717/peerj.2986#supplemental-information.

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
