# Peer review of "The preventive effect of dexmedetomidine on paroxysmal sympathetic hyperactivity in severe traumatic brain injury patients who have undergone surgery: a retrospective study"

_PeerJ, doi:10.7717/peerj.2986_

## Round 0.1 · original submission · Major Revisions

This work maybe suitable for publication after revision according to the reviewer comments

Reviewer 1 ·

Basic reporting

1. There are some minor issues regarding use of the English language. See following annotations:
- Line 18, 30, 59 and 167: “prognosis” (singular) instead of “prognoses”
- Line 68: “routine postoperative sedation” would be a more adequate expression than “the routinely postoperative sedation”
- Line 90: “For the dexmedetomidine group, when each patient was placed in the NICU after his or her operation, dexmedetomidine was administered at an initial loading dose of 0,8 µg/kg within 10 minutes, …”
- Line 97: “.. despite no non-routine procedures were used.”
- Line 107: “by telephone” instead of “over the telephone”
- Line 108: “provided”
- Line 126: “regarding the” instead of “in relation to the”
- Line 140: “The statistical power for the outcome measures described above is relatively…”
- Line 149: “Considering differential diagnoses such as…. is crucial…”
- Line 155: “.. and related theories have not been …”
- Line 157: “However, later studies identified…”
- Line 199: Maybe “statistically significant difference” instead of “statistical difference”?
- Line 206: “This” instead of “These”
- Line 209: “…make a timely diagnosis…”
2. Although widely used, Glasgow outcome scale (GOS) should be backed up by a literature reference (line 29).
3. In line 52, the authors declare that 7,7-33% of patients in ICUs suffer from PSH. It should be made clear whether this statement is true for all kinds of ICU patients or limited to i. e. trauma patients or to a cohort of surgical or even neurosurgical patients.
4. In Line 220, the authors mention “other differences” they did not find without specifying what exactly they meant to find. This paragraph could be written more clearly.
5. Figures 1 and 2 would, in my opinion, require a more precise description of what can be seen in them, for example by altering the title. A possible suggestion would be “Ratio of patients meeting different criteria regarding likelihood of PSH diagnosis”
6. For those cases in which a significant difference was found and depicted in the figures, p-values for those differences should be included in the figures/legend.

Experimental design

1. In line 104ff it is mentioned that two components of PSH-AM were combined to estimate diagnosis likelihood of PSH, but there is no further description what those components actually are. Considering the fact that readers might not have the original paper by Baguley et al (2014) at hand, a more detailed description of what this score actually consists of would be desirable.
2. Furthermore, it is stated that kidney and liver failure led to patient exclusion. To make the exclusion criteria clear especially with respect to possible following studies, it would be helpful to include a definition for this, for example by stating it as “kidney failure as defined by laboratory parameter x exceeding value y” (line 77).
3. If possible, it would be desirable to include some more information about the dose (range) of any other sedatives used in the control group (line 92).

Validity of the findings

In my opinion, the data presented doesn’t support the general conclusion, that “dexmedetomidine has a preventive effect on PSH for [all] patients with sTBI”. Limitations of the study, first of all regarding the focus on a cohort of patients with sTBI who underwent surgery and the limited statistical power of the study, should be stated explicitely in the conclusion. Additionally, it could be mentioned that the preventive effect has been detected “as compared to other sedation regimes”.

Additional comments

Dear Authors, thank you for your interesting submission, which I believe will be suitable for publication with some modifications as mentioned in the comments above.

Reviewer 2 ·

Basic reporting

Very interesting work with a view about an import issue for brain injury patients.
Also good transparency with speaking about limitations

Including criteria is a bit difficult cause some diseases like bradykardia, hypotension and other cardial disorders are contraindicated for dexmedetomidine

What was the reason to evaluate only operated patients?

Experimental design

Initial GOS has to bee reported to compare, also evaluating this important parameter via asking the relatives by call is a bias

Have there been no other medicaments used for sedation respectively preventing paroxysmal sympathetic hyperactivity in control goup?

Descriptive statistic should be written in results part.

Results about death percentage are missing

Validity of the findings

Multivariant analysis between different diagnosis, age, type of surgery would be very interesting. Maybe here you can find statistical significance

These reports show low statistical significance in general

Annotated reviews are not available for download in order to protect the identity of reviewers who chose to remain anonymous.

---

## Round 0.2 · Minor Revisions

The authors have in general adequately revised the ms according to the reviewer comments. Though there are still some minor revisions to undertake, in order to further consider this work for publication in PeerJ. Looking forward to seeing the revised version soon.

Reviewer 1 ·

Basic reporting

I still hold some minor concerns regarding the use of the English language:

line 22: ... were referred to as ...
line 23: ... were referred to as ..
line 59: ... without signifcant inhibition of respiration..
line 73: ... death occuring within ...
line 130: ... had the same mortality ..

Experimental design

no comment

Validity of the findings

no comment

Additional comments

All concerns expressed in my first review were successfully adressed by the authors. I believe the revised manuscript to be suitable for publication after some minor adjustments regarding the use of the English language as listed above.

Reviewer 2 ·

Basic reporting

If possible tables should be matched together for a better and clear illustration

Experimental design

Methods description shows some limitation, for example the dosage and day of sedation are missed

Validity of the findings

p Values are also used for data like gender or number of patients which have obviously no statistical significance.

---

## Round 0.3 · accepted · Accept

After adequately revising the ms according to the reviewer comments, this ms is now ready to be published in PeerJ.